# A comparison of health care worker-collected foam and polyester nasal swabs in convalescent COVID-19 patients

**Brian Hart** [1]*, **Yuan-Po Tu**[2], **Rachel Jennings**[3], **Prateek Verma**[1], **Leah R. Padgett**[4], **Douglas Rains**[4], **Deneen Vojta**[1], **Ethan M. Berke**[1]

**1** Research and Development, UnitedHealth Group, Minnetonka, MN, United States of America, **2** The Everett Clinic, Everett, WA, United States of America, **3** Applied Research Associates, Albuquerque, NM, United States of America, **4** Quantigen Biosciences, Fishers, IN, United States of America

\* brianhart@uhg.com

**Data Availability Statement:** The data cannot be made publicly available due to patient privacy concerns arising from the small sample size of patients collected in a specific timeframe and

## Abstract

Both polyester and foam nasal swabs were collected from convalescent COVID-19 patients at a single visit and stored in viral transport media (VTM), saline or dry. Sensitivity of each swab material and media combination were estimated, three by three tables were constructed to measure polyester and foam concordance, and cycle threshold (Ct) values were compared. 126 visits had polyester and foam swabs stored in viral transport media (VTM), 51 had swabs stored in saline, and 63 had a foam swab in VTM and a polyester swab stored in a dry tube. Polyester and foam swabs had an estimated sensitivity of 87.3% and 94.5% respectively in VTM, 87.5% and 93.8% respectively in saline, and 75.0% and 90.6% respectively for dry polyester and foam VTM. Polyester and foam Ct values were correlated, but polyester showed decreased performance for cases with a viral load near the detection threshold and higher Ct values on average.

## Introduction

In the months since the Centers for Disease Control and Prevention began tracking cases of coronavirus disease 2019 (COVID-19), more than 6.5 million people have officially tested positive in the United States alone, resulting in more than 193,000 deaths [1]. The exponential spread of SARS-CoV-2 has resulted in an enormous demand for testing for the SARS-CoV-2 virus that causes the disease. This increased demand has put a strain on every level of the system from personal protective equipment (PPE) worn by health care workers while administering the test, to swabs for collecting the samples, to viral transport media (VTM) to store the sample until tested, to laboratory capacity [2–4]. Efforts to provide viable alternatives for each of these supply shortages are crucial to keeping the healthcare workforce safe and the testing throughput high.

The shortages of PPE and swabs are driven, in part, because the originally recommended nasopharyngeal (NP) test cannot be self-administered, requiring a change in PPE for each test. A recent study showed that patient-collected foam nasal swabs were comparable to health care worker-collected NP swabs for collecting SARS-CoV-2 virus, providing a safer and less

region of the country. Deidentified data will still be shared upon reasonable request by contacting the UnitedHealth Group Office of Human Research affairs at ohra_uhg@uhg.com or Tracy Ziolek at Tracy_Ziolek@uhg.com. All legal and ethical questions regarding the sharing of this data can also be directed to the same sources.

**Funding:** Laboratory testing was conducted with financial support from Thermo Fisher Scientific. Brian Hart, Rachel Jennings, Prateek Verma, Deneen Vojta, and Ethan Berke were employees of UnitedHealth Group during the design and analysis of the study and initial drafting of the manuscript. Yuan-Po Tu is an employee of The Everett Clinic, which is a subsidiary of UnitedHealth Group. Leah Padgett and Douglas Rains are employees of Quantigen Biosciences and have performed contract services for Thermo Fisher Scientific. UnitedHealth Group, The Everett Clinic, and Quantigen Biosciences provided support in the form of salaries for the authors, but did not have any additional role in the study design, data collection and analysis, decision to publish, or preparation of the manuscript. The specific roles of these authors are articulated in the 'author contributions' section.

**Competing interests:** Brian Hart, Rachel Jennings, Prateek Verma, Deneen Vojta, and Ethan Berke were employees of UnitedHealth Group during the design and analysis of the study and initial drafting of the manuscript. Yuan-Po Tu is an employee of The Everett Clinic, which is a subsidiary of UnitedHealth Group. Leah R. Padgett and Douglas Rains are employees of Quantigen Biosciences and have performed contract services for Thermo Fisher Scientific. The specific roles of these authors are articulated in the 'author contributions' section. The competing interests implied by the authors' employer/employee relationships do not alter our adherence to PLOS ONE policies on sharing data and materials.

invasive sampling method [5]. Similarly, another independent study demonstrated that patient-collected nasal swabs were comparable to health care worker-collected oropharyngeal or nasal swabs [6]. These results validate findings for influenza testing [7, 8]. While these results open the possibility of using patient-collected foam nasal swabs for sample collection, reducing the risk of viral exposure for health care workers, several issues remain. In the United States, foam nasal and nylon flocked swabs are not as readily available or mass produced as polyester swabs [9]. Additionally, swabs are typically stored and transported in VTM under refrigeration at or below 4°C. With VTM and swab supplies running low, and difficulties in obtaining sufficient refrigeration space for the massive number of samples arriving at the labs to be tested, testing requirements must be reevaluated to see if they can be safely altered [10, 11]. Recent work has shown that saline may be a suitable replacement for VTM [12]. Since these findings were released, the Food and Drug Administration (FDA) has updated their testing recommendations to allow for a wider variety of substances to be used for viral transport and stabilization and allow for self-collected foam nasal swabs when VTM and NP swabs are not available [13].

To address the dwindling supply of recommended swabs, we compared the relative performance of polyester and foam nasal swabs for collecting SARS-CoV-2, stored and transported either in VTM, saline, or in a dry tube.

## Methods

### Population and sample collection

Patients who have previously tested positive for SARS-CoV-2 at any site from the original study in Washington state were approached to return for additional testing [5]. A cohort of 63 positive patients returned 7–9, 14–18, and 28–31 days after their initial positive diagnosis for longitudinal sample collection, with swabbing continuing until all results for samples from a single visit were negative. A second cohort consisted of participants who also previously tested positive. They were contacted by ambulatory clinic staff and asked to return for a single visit as soon as possible.

People who agreed to participate in either study were consented by medical staff. Participants were evaluated at a designated ambulatory clinic site that only saw patients that were confirmed with a prior positive SARS-CoV-2 reverse transcription polymerase chain reaction (RT-PCR) test. Inclusion criteria included a previous positive SARS-CoV-2 test and the ability to consent and agree to participate in the study. People who were not able to demonstrate understanding of the study, not willing to commit to having all samples collected, had a history of nosebleed in the past 24 hours, nasal surgery in the past two weeks, chemotherapy treatment with documented low platelet and low white blood cell counts, or acute facial trauma were excluded from the study. Health care workers used a written consent form to explain the study and give eligible patients the opportunity to decline. This study protocol was deemed to be part of a minimal risk research protocol by the Office of Human Research Affairs at United-Health Group.

In the first cohort of 63 SARS-CoV-2 positive individuals, three nasal swabs were collected by a health care worker from each patient: a foam swab and two polyester swabs. The first swab was gently inserted into the right nostril until resistance was met at the level of the turbinate (less than one inch into the nostril) and gentle pressure was applied to the outside nasal wall and the swab was rotated several times against the nasal wall and then slowly removed from the nostril. The second swab was then gently inserted into the left nostril and sampling was obtain in a similar manner. Next, the first swab was inserted into the left nostril and

sampling was obtained in a similar manner. Finally, the second swab was inserted into the right nostril and sampling was obtained in a similar manner.

Left and right nostril randomization was based on the patient's year of birth. For patient born in an odd numbered year, the left nostril was initially sampled with the foam swab followed by their right nostril being sampled by the polyester swab. Conversely, for patients born in an even numbered year, the right nostril was initially sampled by the foam swab followed by their left nostril being sampled by the polyester swab. The foam and polyester swabs were eluted in viral transport media, stored at 4˚C, and sent to a reference laboratory for immediate testing.

After obtaining and storing the foam and first polyester swabs, a second polyester swab was used to sample both the right and left nostril in a similar manner to the previous swabs. The second polyester swab was then inserted in a dry tube (10 mL BD Vacutainer), stored at 4˚C, and sent to a reference laboratory for immediate testing. Note that this sample collection procedure put the polyester swab stored in the dry tube at a disadvantage as it was always the third swab in a nostril and thus potentially had less virus available to sample. Due to participant testing constraints, the polyester swab to be stored in a dry tube was not collected from all patients after the first visit. The three swabs will be referred to as the foam, VTM polyester, and dry polyester swabs.

Patients in the second cohort were also convalescing from COVID-19. They were recruited immediately after their initial positive test result for SARS-CoV-2. During a single visit (between two and seven days after initial diagnosis), these patients were swabbed twice: once with a polyester swab and once with a foam swab. The order of sample collection was randomized and collected in the same manner as the first two swabs in the first cohort. Both samples were stored at 4˚C and immediately shipped to the reference laboratory for testing. All nasal swab samples were collected by health care workers and marked with a patient study number, swab type, and media type (if any).

RT-PCR was performed by a reference laboratory using the TaqPathTM COVID-19 Combo Kit (ThermoFisher; A47814). Cycle threshold (Ct) values for all samples were reported back to the clinical sites based on three different RNA targets: S Gene, ORF1ab, and N Gene. A higher Ct value corresponds to a lower viral load. Per the reference laboratory standard protocols, a sample was considered positive if two or three Ct values were less than or equal to 37, negative if all three Ct values were greater than 37, and inconclusive otherwise. All inconclusive samples were retested and the results of the retest were used, per manufacturer protocol. While a Ct value of 37 was used to determine the clinical results of the test, RT-PCR continued until 40 cycles were completed. Therefore, the maximum Ct value in the data is 40. Foam swabs were stored in 1 mL of VTM or 2 mL of saline and vortexed for three to five seconds. Polyester swabs were stored in 3 mL of VTM or 2 mL of saline and vortexed for three to five seconds. Dry polyester swabs were eluted in 1 mL of phosphate-buffered saline (PBS) and vortexed for 30 seconds followed by a ten-minute incubation at room temperature.

Because of the volumetric differences among the three types of collection and/or elution media, mathematical adjustments to some Ct values were made in order to account for the differences in dilution of the various swab samples. By scaling (i.e., normalizing) Cts for both the 2 mL saline samples and the 3 mL VTM samples to match the 1 mL sample volume (VTM or 1 mL PBS dry swab elution), the sensitivity comparisons among the three sample collection volumes are more legitimate than using unnormalized data. Values were normalized as follows: to account for the 2-fold dilution difference between saline (2 mL) and dry polyester (1 mL PBS) or foam (1 mL VTM), 1 Ct was subtracted from all qPCR results for the former; similarly, to account for the 3-fold sample volume difference between the 3 mL VTM polyester swab samples and dry polyester (1 mL PBS) or foam (1 mL VTM), 1.585 Ct was subtracted from the

former. Note that the specific Ct adjustments follow the formula $2^{-dCt} = F$, where dCt is the necessary Ct adjustment and F is the fold increase in volume. This formula assumes the assays perform at approximately 100% PCR efficiency during the geometric phase of amplification [14].

## Statistical analysis

We report the 3x3 confusion matrix showing the results of the polyester swabs against the results of the foam swabs. We estimate the sensitivity of the polyester and foam swabs, considering a patient to be a true positive if either swab returned a positive result and negative otherwise, and report the 95% confidence intervals for the sensitivity estimates. Note that this definition treats all positives as true positives and, thus, implies a specificity of 100% for all swabs. While this is a strong assumption, it allows for the estimation of the sensitivity of each swab without assuming one swab type is a gold standard truth. Confusion matrices and sensitivity estimates are broken out by visit number where applicable. Fisher's exact tests are used to calculate p-values to compare the sensitivity of foam and polyester swabs in different media types. P-values were not calculated for individual visits because of small sample sizes.

For each RNA target, we calculated the correlation of the normalized Ct values between the foam and polyester swabs for all pairs where at least one of the Ct values was $< 40$, indicating enough of the RNA target was detected to stop the RT-PCR machine short of the maximum Ct value. For the same subset of polyester/foam swab pairs with at least one Ct $< 40$ we also took the difference between the two Ct values and plotted the results in a box plot. Positive difference values indicate a higher Ct (and thus lower viral load) in the polyester samples and negative values indicate a higher Ct in the foam samples. All statistical analysis was performed using R version 3.6.1 [15].

## Results

Of the 63 patients in the first cohort, 29 were positive at the 7–9 day visit (Visit 1), five were positive at the 14–18 day visit (Visit 2), and zero were positive at the 28–31 day visit (Visit 3). One patient's swabs from the 28–31 day visit leaked in transport and were not tested. The second cohort consisted of 76 participants, of which 51 had their swab samples stored in saline and 25 had their swab samples stored in VTM. A summary of the patient demographics in the two groups can be found in Table 1.

A summary of the demographics of the two cohorts. Cohort 1 consists of 63 patients who returned for follow-up visits until all swabs tested negative for up to three visits. Cohort 2 consists of 71 patients with a single visit each. All dry polyester swabs were collected from Cohort 1. All saline swabs were collected from Cohort 2. VTM swabs were collected from both cohorts.

**Table 1. Cohort demographics.**

|  | Cohort 1 (N = 63) | Cohort 2 (N = 76) |
| --- | --- | --- |
| Age, median (IQR) | 46 (36, 56.5) | 45 (34.75, 51.25) |
| Gender |  |  |
| Female, n (%) | 34 (54.0%) | 36 (47.3%) |
| Male, n (%) | 29 (46.0%) | 40 (52.6%) |
| Days since diagnosis, mean (sd) | 13.8 (5.1) | 10.0 (6.4) |
| Days since first symptoms, mean (sd) | 7.8 (1.6) | 3.7 (3.1) |

**Table 2. VTM results.**

| | | | VTM Polyester | | | | Polyester Sens. (95% CI) | Foam Sens. (95% CI) |
|---|---|---|---|---|---|---|---|---|
| | Visit 1 (7–9 Days) | | Positive | Negative | Inconclusive | Total | | |
| | | Positive | 42 | 5 | 0 | 47 | 45/50 | 47/50 |
| | | Negative | 3 | 32 | 0 | 35 | 90.0% | 94.0% |
| | | Inconclusive | 0 | 6 | 0 | 6 | (77.4%, 96.2%) | (82.5%, 98.4%) |
| | | Total | 45 | 43 | 0 | 88 | | |
| | Visit 2 (14–18 Days) | | Positive | Negative | Inconclusive | Total | | |
| | | Positive | 3 | 1 | 1 | 5 | 3/5 | 5/5 |
| VTM Foam | | Negative | 0 | 27 | 0 | 27 | 60.0% | 100% |
| | | Inconclusive | 0 | 2 | 0 | 2 | (17.0%, 92.7%) | (46.2%, 100%) |
| | | Total | 3 | 30 | 1 | 34 | | |
| | Visit 3 (28–31 Days) | | Positive | Negative | Inconclusive | Total | | |
| | | Positive | 0 | 0 | 0 | 0 | | |
| | | Negative | 0 | 4 | 0 | 4 | NA | NA |
| | | Inconclusive | 0 | 0 | 0 | 0 | | |
| | | Total | 0 | 4 | 0 | 4 | | |
| | All Time Points | | Positive | Negative | Inconclusive | Total | | |
| | | Positive | 45 | 6 | 1 | 52 | 48/55 | 52/55 |
| | | Negative | 3 | 63 | 0 | 66 | 87.3% | 94.5% |
| | | Inconclusive | 0 | 8 | 0 | 8 | (74.9%, 94.3%) | (83.9%, 98.6%) |
| | | Total | 48 | 77 | 1 | 126 | p-value: 0.32 | |

A 3x3 table and estimated sensitivities for each visit and all visits combined of the test results for the foam and polyester nasal samples, both stored in VTM.

Tests run on the polyester nasal swabs detected four fewer positive cases than foam swabs and had an estimated sensitivity (95% confidence interval) of 87.3% (74.9%, 94.3%) when stored in VTM compared to 94.5% (83.9%, 98.6%) for the foam swabs in VTM (p-value: 0.32, Table 2). Additionally, there were eight visits with an inconclusive VTM foam swab and a negative VTM polyester swab, but no visits with an inconclusive VTM polyester swab and a negative VTM foam swab. In saline, tests run on the polyester swabs detected two fewer positive cases and had an estimated sensitivity of 87.5% (70.1%, 95.9%) compared to 93.8% (77.8%, 98.9%) for foam (p-value: 0.67, Table 3). Tests run on the dry polyester swabs detected five fewer cases than foam swabs in VTM and had an estimated sensitivity of 75.0% (56.2%, 87.9%) compared to 90.6% (73.8% to 97.5%) for VTM foam from the same visits (p-value: 0.18, Table 4).

**Table 3. Saline results.**

| | | Saline Polyester | | | | Polyester Sens. (95% CI) | Foam Sens. (95% CI) |
|---|---|---|---|---|---|---|---|
| | | Positive | Negative | Inconclusive | Total | | |
| Saline Foam | Positive | 26 | 4 | 0 | 30 | 28/32 | 30/32 |
| | Negative | 2 | 17 | 1 | 20 | 87.5% | 93.8% |
| | Inconclusive | 0 | 1 | 0 | 1 | (70.1%, 95.9%) | (77.8%, 98.9%) |
| | Total | 28 | 22 | 1 | 51 | p-value: 0.67 | |

A 3x3 table of the test results and estimated sensitivities with 95% confidence intervals for the foam and polyester nasal samples, both stored in saline. Note that the saline results were from a cohort with a single visit and are, thus, not broken out by visit.

**Table 4. Dry results.**

| | | Dry Polyester | | | | Polyester Sens. (95% CI) | Foam Sens. (95% CI) |
|---|---|---|---|---|---|---|---|
| | | Positive | Negative | Inconclusive | Total | | |
| **VTM Foam** | Positive | 21 | 5 | 3 | 29 | 24/32 | 29/32 |
| | Negative | 2 | 26 | 2 | 30 | 75.0% | 90.6% |
| | Inconclusive | 1 | 3 | 0 | 4 | (56.2%, 87.9%) | (73.8%, 97.5%) |
| | Total | 24 | 34 | 5 | 63 | p-value: 0.18 | |

A 3x3 table and estimated sensitivities with 95% confidence intervals for the VTM foam and dry polyester nasal samples. Note that the dry swabs were only collected at the first visit, so results are not broken out by visit.

Correlations between foam and polyester Ct values were highest in the saline samples, followed by the VTM samples and then the dry polyester versus VTM foam samples (Fig 1). While Ct correlations were highest in saline, the foam Ct value was lower than the polyester Ct value more than 63% of the time for the saline swabs, indicating the increased viral loads estimated using foam swabs (Fig 2). Similar biases toward lower Ct values appeared in the VTM polyester versus VTM foam and dry polyester versus VTM foam comparisons. The tendency toward lower Ct values for foam swabs is also apparent in the paired Ct plots (Fig 3). Fig 3 is further broken out by samples taken less than ten days from symptom onset and samples taken at least 10 days from symptom onset in S1 and S2 Figs.

## Discussion

Direct comparison of foam and polyester nasal swabs stored in VTM, in saline, or dry demonstrated decreased, but potentially adequate performance of polyester swabs in times of shortages. The estimated sensitivity of the polyester swabs was 87.3% and 87.5% in VTM and saline respectively compared to 94.5% and 93.8% for foam swabs. These calculations were

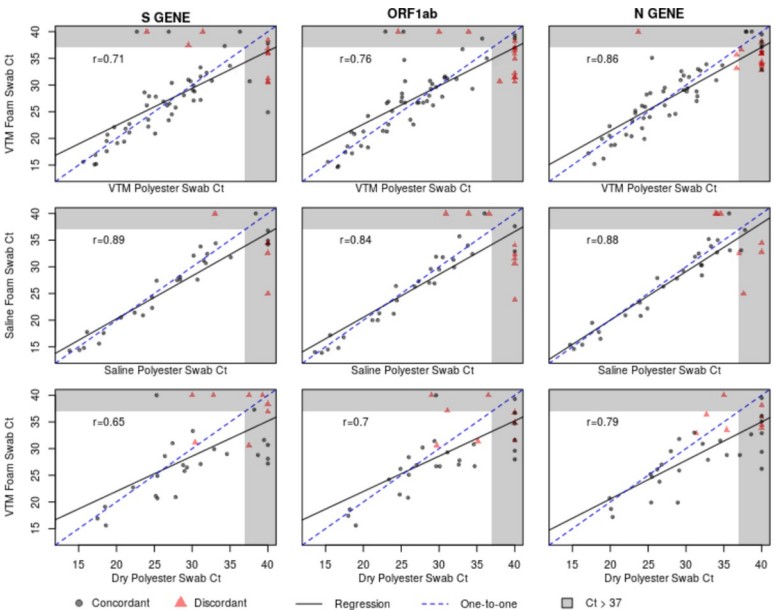

**Fig 1. Ct correlation plots.** Plots showing the cycle threshold (Ct) values for each of the three RNA targets and three transport media. The black line represents the best fitting linear regression, the dashed blue line represents a perfect one-to-one relationship.

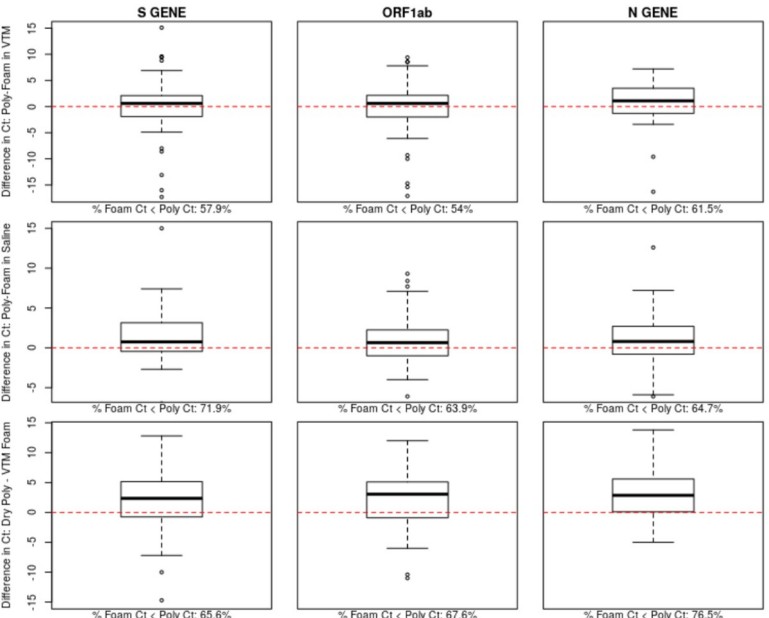

**Fig 2. Ct difference boxplots.** Plots showing the difference in cycle threshold (Ct) of the polyester and foam swabs collected at the same visits. Positive values represent higher Ct values in the polyester swab. The dashed red line represents equivalent Ct values. The percentage of samples for which the foam swab has a lower Ct value is shown below each sub-plot.

conservative because they counted inconclusive polyester results as negatives for the purposes of sensitivity estimation, although no positive VTM foam swabs had a corresponding inconclusive VTM polyester swab. The assumption used for sensitivity estimation that any positive result was a true positive is reasonable when viewed in the light that all patients had previously tested positive for SARS-CoV-2.

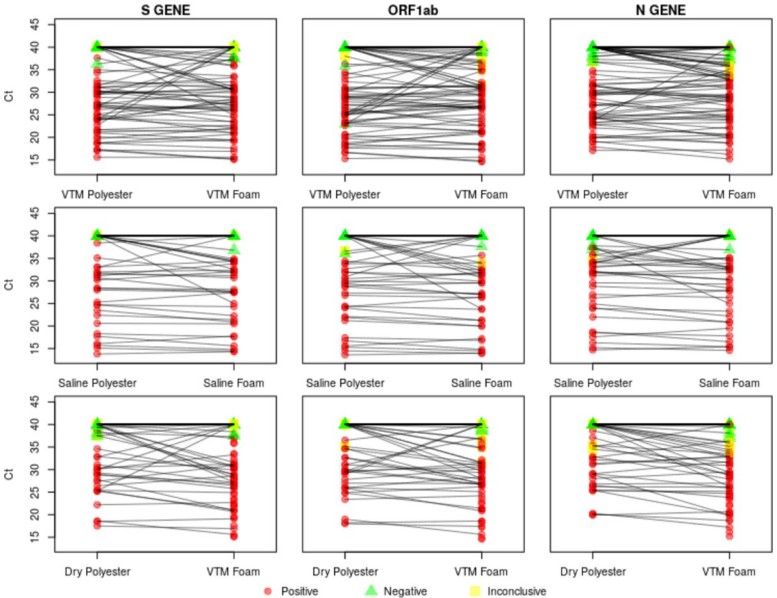

**Fig 3. Paired Ct plots.** Paired Ct plots showing the polyester and foam Ct values for each transport media and target gene combination considered. Swabs collected at the same visit are connected by a black line.

The comparison of polyester and foam swabs did not differ between saline and VTM storage. While the estimated sensitivity above 87% may be deemed sufficient in times of a public health emergency, the strength of the findings is diminished for cases with a viral load near the positive/negative threshold of 37 cycles. This issue can be most clearly seen via the cluster of points along the upper edge of the right-hand side of the correlation plots. These points represent cases where tests run on foam swabs detected virus and the polyester did not. This finding is also reflected in the eight previously mentioned inconclusive VTM foam results that were negative for the VTM polyester swab. All three polyester versus foam swab comparisons exhibited a tendency for the foam swab to have lower Ct values than the polyester swab, indicating the foam swab's superior ability to collect virus. Despite these reductions in performance, polyester swabs in VTM or saline may be an adequate sample collection method in cases where foam nasal and NP swabs have been entirely exhausted, a situation which exists in many locations.

While the dry polyester swabs appeared to show poor performance, they were put at a disadvantage by the study design. Because all dry polyester swabs were the third swab collecting a sample from a nostril at a given visit, the decrease in performance cannot be attributed solely to the swab type and lack of storage media. Dry swabs were only collected for the first follow-up visit in the cohort of 63 patients that also had swabs stored in VTM, resulting in a smaller sample size and different time since symptom onset than the other comparisons. Dry swabs have been found to have high sensitivity in other diagnostic settings and should not be ruled out entirely based on this study [16].

The current study has several limitations. All participants were convalescing COVID-19 patients, and the time from first symptoms and first diagnosis to test date varied from test to test. As patients progressed further from their diagnosis date their viral loads dropped, creating more cases where the Ct values were near the border of detection. Other research has suggested viral load may have already peaked at the time of diagnosis followed by a slow decline over time [17]. The samples from this study may not be representative of testing in newly infected patients who are seeking their first SARS-CoV-2 test. Additionally, because no NP swabs were obtained for this study, performance of the polyester swab cannot be directly compared to the FDA's preferred swabbing method [13]. Although final Ct values were adjusted for varying amounts of transport media, imprecision in these adjustments could confound data interpretation; specifically, comparisons between the mathematically corrected samples (2 mL and 3 mL) and the non-corrected samples (1 mL) present the greatest risk for error. Finally, the study was not designed to compare the performance of VTM and saline or to estimate an interaction of swab type and transport media type.

Despite these limitations, polyester swabs stored in VTM or saline may be a viable sample collection method for COVID-19 testing, especially in light of the shortages of other swab types. The viability of polyester swabs is most clearly demonstrated via the high correlation between polyester and foam Ct values from the same visit. Any recommendation for polyester swab usage should bear in mind that the decrease in performance near the border of detection may lead to false negatives in patients with low viral loads.

## Supporting information

**S1 Fig. Paired Ct plots: Newly symptomatic.** Paired Ct plots showing the polyester and foam Ct values for each transport media and target gene combination considered for swabs from patients less than 10 days from symptom onset at the time of sample collection. Swabs collected at the same visit are connected by a black line.
(TIF)

**S2 Fig. Paired Ct plots: Not newly symptomatic.** Paired Ct plots showing the polyester and foam Ct values for each transport media and target gene combination considered for swabs from patients at least 10 days from symptom onset at the time of sample collection. Swabs collected at the same visit are connected by a black line.
(TIF)

## Acknowledgments

The authors acknowledge the contributions of James S Elliott and Lauren A. Kennington from Quantigen Biosciences, Lindsay Nelson from UnitedHealth Group, and Garrett Galbreath, the health care workers, and staff from The Everett Clinic.

## Author Contributions

**Conceptualization:** Brian Hart, Yuan-Po Tu, Rachel Jennings, Prateek Verma, Deneen Vojta, Ethan M. Berke.

**Data curation:** Brian Hart, Leah R. Padgett, Douglas Rains, Ethan M. Berke.

**Formal analysis:** Brian Hart, Rachel Jennings, Leah R. Padgett, Douglas Rains.

**Funding acquisition:** Prateek Verma, Deneen Vojta.

**Investigation:** Yuan-Po Tu, Leah R. Padgett, Douglas Rains, Deneen Vojta, Ethan M. Berke.

**Methodology:** Yuan-Po Tu, Rachel Jennings, Prateek Verma, Deneen Vojta, Ethan M. Berke.

**Project administration:** Prateek Verma, Deneen Vojta.

**Supervision:** Yuan-Po Tu, Rachel Jennings, Deneen Vojta, Ethan M. Berke.

**Validation:** Yuan-Po Tu, Rachel Jennings.

**Visualization:** Brian Hart.

**Writing – original draft:** Brian Hart.

**Writing – review & editing:** Brian Hart, Yuan-Po Tu, Rachel Jennings, Prateek Verma, Leah R. Padgett, Douglas Rains, Deneen Vojta, Ethan M. Berke.

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
