## [Decision Letter · Decision Letter 0]

2 Sep 2020

PONE-D-20-13951

A comparison of health care worker-collected foam and polyester nasal swabs in convalescent COVID-19 patients

PLOS ONE

Dear Dr. Hart,

Thank you for submitting your manuscript to PLOS ONE. After careful consideration, we feel that it has merit but does not fully meet PLOS ONE’s publication criteria as it currently stands. Therefore, we invite you to submit a revised version of the manuscript that addresses the points raised during the review process.

The Authors are expected to address all the criticisms by all Reviewers. In particular, please discuss if different volume of VTM or saline used, or normalization of Ct values would affect the results (Reviewer #1) and provide basic clinical information of the two cohorts (Reviewer #2). In additional to the above comments, please address,

Please present the number of samples collected by day of visit groupTables 1-3 have pooled samples collected at different days after diagnosis. However, viral shedding decreased over time and hence test sensitivity is unlikely constant over sampling dates. Please further assess the sensitivity by the time of sampling.The authors have assessed sensitivity of different sampling methods. Could the authors explain why specificity was not assessed?

We look forward to receiving your revised manuscript.

Kind regards,

Eric HY Lau, Ph.D.

Academic Editor

PLOS ONE

Additional Editor Comments:

The Authors are expected to address all the criticisms by all Reviewers. In particular, please discuss if different volume of VTM or saline used, or normalization of Ct values would affect the results (Reviewer #1) and provide basic clinical information of the two cohorts (Reviewer #2). In additional to the above comments, please address,

1. Please present the number of samples collected by day of visit group

2. Tables 1-3 have pooled samples collected at different days after diagnosis. However, viral shedding decreased over time and hence test sensitivity is unlikely constant over sampling dates. Please further assess the sensitivity by the time of sampling.

3. The authors have assessed sensitivity of different sampling methods. Could the authors explain why specificity was not assessed?

Journal Requirements:

"Laboratory testing was conducted with financial support from Thermo Fisher Scientific."

We note that one or more of the authors are employed by a commercial company: UnitedHealth Group and Quantigen Biosciences.

2.1. Please provide an amended Funding Statement declaring this commercial affiliation, as well as a statement regarding the Role of Funders in your study. If the funding organization did not play a role in the study design, data collection and analysis, decision to publish, or preparation of the manuscript and only provided financial support in the form of authors' salaries and/or research materials, please review your statements relating to the author contributions, and ensure you have specifically and accurately indicated the role(s) that these authors had in your study. You can update author roles in the Author Contributions section of the online submission form.

2.2. Please also provide an updated Competing Interests Statement declaring this commercial affiliation along with any other relevant declarations relating to employment, consultancy, patents, products in development, or marketed products, etc.  

3. We noted in your submission details that a portion of your manuscript may have been presented or published elsewhere.

"A preprint of the manuscript has been posted on medRxiv, but the manuscript has not other wise been published. " ext-link-type="uri" xlink:type="simple">https://www.medrxiv.org/content/10.1101/2020.04.28.20083055v1"

Please clarify whether this  publication was peer-reviewed and formally published. If this work was previously peer-reviewed and published, in the cover letter please provide the reason that this work does not constitute dual publication and should be included in the current manuscript.

Reviewers' comments:

Reviewer's Responses to Questions

**Comments to the Author**

1. Is the manuscript technically sound, and do the data support the conclusions?

Reviewer #1: Yes

Reviewer #2: Yes

2. Has the statistical analysis been performed appropriately and rigorously? 

Reviewer #1: Yes

Reviewer #2: Yes

3. Have the authors made all data underlying the findings in their manuscript fully available?

Reviewer #1: Yes

Reviewer #2: Yes

4. Is the manuscript presented in an intelligible fashion and written in standard English?

Reviewer #1: Yes

Reviewer #2: Yes

5. Review Comments to the Author

Reviewer #1: A timely trial of polyester vs foam nasal swabs stored and/or tested using VTM, saline, or dry tubes with buffered saline.

Comment only: It is unfortunate that you could not have treated the samples more similarly. I’m not sure why you chose to use different amount of VTM and saline, for instance, at the lab. Correcting for/normalizing the Ct values is adequate but not as convincing.

Did you try to correct for the first/second collection of swabs in the analysis? It is not necessary, just a thought.

Swabs do not detect virus, but rather “collect” them. Consider changing the wording.

Discussion: line 186, please add, “in times of shortages” or something to that effect.

Reviewer #2: In the context of the COVID-19 pandemic, the simple, convenient and effective detection methods are worth promoting. The polyester swabs stored in VTM or saline may be a viable sample collection method for COVID-19.

But in the RT-PCR part, the author should show more details.

In addition, the basic clinical information of the two cohorts should be presented, such as sex, BMI, age, symptoms and so on.

6. PLOS authors have the option to publish the peer review history of their article (what does this mean?). If published, this will include your full peer review and any attached files.

Reviewer #1: No

Reviewer #2: No

---

## [Author Response · Author response to Decision Letter 0]

16 Sep 2020

The authors would like to thank the reviewers and editor for their careful review and consideration of our manuscript titled “A comparison of health care worker-collected foam and polyester nasal swabs in convalescent COVID-19 patients”. Below we provide a point by point response to the reviewer comments below:

• Please discuss if different volume of VTM or saline used, or normalization of Ct values would affect the results (Reviewer #1).

Yes, using different sample volumes and applying mathematical corrections is not ideal and could affect the results. However, considering the urgent need for these studies, the standard 1-3 mL volumes used for clinical sample collection, and the valid mathematical approach for performing fold change calculations on qPCR, we felt normalization of the data was the best approach. For clarification, we have added text to the methods section and to the discussion. 

Methods:

“Because of the volumetric differences among the three types of collection and/or elution media, mathematical adjustments to some Ct values were made in order to account for the differences in dilution of the various swab samples. By scaling (i.e., normalizing) Cts for both the 2 mL saline samples and the 3 mL VTM samples to match the 1 mL sample volume (VTM or 1 mL PBS dry swab elution), the sensitivity comparisons among the three sample collection volumes are more legitimate than using unnormalized data. Values were normalized as follows: to account for the 2-fold dilution difference between saline (2 mL) and dry polyester (1 mL PBS) or foam (1 mL VTM), 1 Ct was subtracted from all qPCR results for the former; similarly, to account for the 3-fold sample volume difference between the 3 mL VTM polyester swab samples and dry polyester (1 mL PBS) or foam (1 mL VTM), 1.585 Ct was subtracted from the former. Note that the specific Ct adjustments follow the formula 2-dCt=F, where dCt is the necessary Ct adjustment and F is the fold increase in volume. This formula assumes the assays perform at approximately 100% PCR efficiency during the geometric phase of amplification.”

Discussion:

“Although final Ct values were adjusted for varying amounts of transport media, imprecision in these adjustments could confound data interpretation; specifically, comparisons between the mathematically corrected samples (2 mL and 3 mL) and the non-corrected samples (1 mL) present the greatest risk for error.” 

• Provide basic clinical information of the two cohorts (Reviewer #2).

Thank you for the suggestion. We have added in a new table one with basic clinical/demographic information available on the two cohorts. 

• Please present the number of samples collected by day of visit group

These values can now be found in the expanded version of Tables 2 (previously Table 1).

• Tables 1-3 have pooled samples collected at different days after diagnosis. However, viral shedding decreased over time and hence test sensitivity is unlikely constant over sampling dates. Please further assess the sensitivity by the time of sampling.

Table 2 (previously called Table 1) has been expanded to breakout the results by the three different visits. Note that Tables 3 and 4 contain results which were collected at a single visit, so no similar table breakout was necessary/possible.

• The authors have assessed sensitivity of different sampling methods. Could the authors explain why specificity was not assessed? 

When estimating sensitivity we assume that all positives were true positives. The primary advantage of this approach is that it allows for the estimation of sensitivity without assuming a single method is a gold standard “truth”. The disadvantage of our approach is that assuming all positives are true positives implies that all methods have a specificity of 100%. One could alternatively treat a single method (e.g. foam VTM) as the truth and compare the other methods when estimating sensitivity. This approach also involves strong assumptions and, while it allows for the estimation of both sensitivity and specificity of the non-reference methods, it implies 100% sensitivity and specificity of the reference method. Based on these tradeoffs, we chose to treat any positive as a true positive even though it precludes accurate specificity estimation.

In addition to the edits made in response to reviewer comments we have made some additional changes to the description of the procedures. These changes were made so that the manuscript accurately describes how the study was carried out in practice. The sample collection procedures are now more accurately and clearly described in the manuscript. We have also changed the method used to calculate the 95% confidence intervals from the Normal approximation to the Wilson Score method, which has been shown to have better coverage. This change was motivated by the need for more stable confidence interval estimates given the low sample sizes, especially when broken out by visit.

---

## [Decision Letter · Decision Letter 1]

8 Oct 2020

PONE-D-20-13951R1

A comparison of health care worker-collected foam and polyester nasal swabs in convalescent COVID-19 patients

PLOS ONE

Dear Dr. Hart,

Thank you for submitting your manuscript to PLOS ONE. After careful consideration, we feel that it has merit but does not fully meet PLOS ONE’s publication criteria as it currently stands. Therefore, we invite you to submit a revised version of the manuscript that addresses the points raised during the review process.

The Authors have addressed most of the reviewers’ comments satisfactorily, but are expected to address the remaining criticisms by Reviewer #1.

We look forward to receiving your revised manuscript.

Kind regards,

Eric HY Lau, Ph.D.

Academic Editor

PLOS ONE

Additional Editor Comments (if provided):

The Authors have addressed most of the reviewers’ comments satisfactorily, but are expected to address the remaining criticisms by Reviewer #1.

Reviewers' comments:

Reviewer's Responses to Questions

**Comments to the Author**

1. If the authors have adequately addressed your comments raised in a previous round of review and you feel that this manuscript is now acceptable for publication, you may indicate that here to bypass the “Comments to the Author” section, enter your conflict of interest statement in the “Confidential to Editor” section, and submit your "Accept" recommendation.

Reviewer #1: (No Response)

Reviewer #2: All comments have been addressed

2. Is the manuscript technically sound, and do the data support the conclusions?

Reviewer #1: Yes

Reviewer #2: Yes

3. Has the statistical analysis been performed appropriately and rigorously? 

Reviewer #1: No

Reviewer #2: Yes

4. Have the authors made all data underlying the findings in their manuscript fully available?

Reviewer #1: No

Reviewer #2: Yes

5. Is the manuscript presented in an intelligible fashion and written in standard English?

Reviewer #1: Yes

Reviewer #2: Yes

6. Review Comments to the Author

Reviewer #1: Thank you for the considerable revision you made. This revision is substantially clearer and more complete.

A few comments:

line 249-250 in the discussion: there is a sentence fragment that needs to be removed or completed.

Although I will not "require" it, it would be helpful to provide p-values for comparisons across swab types and/or storage.

Reviewer #2: 1. The study presents the results of original research.

2. Results reported have not been published elsewhere.

3. Experiments, statistics, and other analyses are performed to a high technical standard and are described in sufficient detail.

4. Conclusions are presented in an appropriate fashion and are supported by the data.

5. The article is presented in an intelligible fashion and is written in standard English.

6. The research meets all applicable standards for the ethics of experimentation and research integrity.

7. The article adheres to appropriate reporting guidelines and community standards for data availability.

7. PLOS authors have the option to publish the peer review history of their article (what does this mean?). If published, this will include your full peer review and any attached files.

Reviewer #1: No

Reviewer #2: No

---

## [Author Response · Author response to Decision Letter 1]

8 Oct 2020

The authors would like to thank the reviewers and editor for their careful review and consideration of our manuscript titled “A comparison of health care worker-collected foam and polyester nasal swabs in convalescent COVID-19 patients”. Below we provide a point by point response to the reviewer comments below:

• line 249-250 in the discussion: there is a sentence fragment that needs to be removed or completed.

Thank you for catching the error. The sentence fragment has been removed in the resubmitted manuscript. 

• Although I will not "require" it, it would be helpful to provide p-values for comparisons across swab types and/or storage.

We have now provided p-values from a Fisher’s Exact test for the three polyester versus foam sensitivity comparisons. The p-values are included in the main text and in Tables 2-4. In Table 2, p-values were not included for the visit specific sensitivity comparisons due to small sample sizes. Two sentences have been added to the Statistical analysis section to explain methods used for p-value calculations.

No additional changes were made to the manuscript. We again thank the reviewers for their constructive feedback which has led to a much improved manuscript.

---

## [Editor Report · Decision Letter 2]

9 Oct 2020

A comparison of health care worker-collected foam and polyester nasal swabs in convalescent COVID-19 patients

PONE-D-20-13951R2

Dear Dr. Hart,

We’re pleased to inform you that your manuscript has been judged scientifically suitable for publication and will be formally accepted for publication once it meets all outstanding technical requirements.

Kind regards,

Eric HY Lau, Ph.D.

Academic Editor

PLOS ONE
---

## [Editor Report · Acceptance letter]

16 Oct 2020

PONE-D-20-13951R2 

A comparison of health care worker-collected foam and polyester nasal swabs in convalescent COVID-19 patients 

Dear Dr. Hart:

I'm pleased to inform you that your manuscript has been deemed suitable for publication in PLOS ONE. Congratulations! Your manuscript is now with our production department. 

Kind regards, 

on behalf of

Dr. Eric HY Lau 

Academic Editor

PLOS ONE